# The Caspase-3 homolog DrICE regulates endocytic trafficking during *Drosophila* tracheal morphogenesis

Saoirse S. McSharry [1] & Greg J. Beitel [1]

Although well known for its role in apoptosis, the executioner caspase DrICE has a non-apoptotic function that is required for elongation of the epithelial tubes of the *Drosophila* tracheal system. Here, we show that DrICE acts downstream of the Hippo Network to regulate endocytic trafficking of at least four cell polarity, cell junction and apical extracellular matrix proteins involved in tracheal tube size control: Crumbs, Uninflatable, Kune-Kune and Serpentine. We further show that tracheal cells are competent to undergo apoptosis, even though developmentally-regulated DrICE function rarely kills tracheal cells. Our results reveal a developmental role for caspases, a pool of DrICE that co-localizes with Clathrin, and a mechanism by which the Hippo Network controls endocytic trafficking. Given reports of in vitro regulation of endocytosis by mammalian caspases during apoptosis, we propose that caspase-mediated regulation of endocytic trafficking is an evolutionarily conserved function of caspases that can be deployed during morphogenesis.

[1] Department of Molecular Biosciences, Northwestern University, Evanston, IL 60208, USA. Correspondence and requests for materials should be addressed to G.J.B. (email: beitel@northwestern.edu)

pithelial tubes of precise sizes are essential for gas exchange and nutrient delivery in animal tissues. Failure of correct tube sizing can lead to fatal disease[1,2], yet the cellular and molecular mechanisms that regulate tube size remain poorly understood. To uncover these mechanisms, we study the tracheal system of the *Drosophila* embryo, a ramifying tubular network that serves as the fly's combined pulmonary and vascular systems[3].

The diameter of the largest tube in the tracheal system, the dorsal trunk (DT), increases twofold as length increases ~15% over a 2.5 h period during mid-embryogenesis, and it does so with no accompanying changes in cell number[4]. Instead, DT dimensions are regulated by a complex set of interacting pathways that all rely on the endocytic system. An apical/lumenal extracellular matrix (aECM) that restricts elongation depends on regulated secretion and endocytosis of matrix-modifying proteins such as Serpentine (Serp)[5]. Basolateral cell-junctional complexes called Septate Junctions (SJs) also restrict DT dimensions. SJs contain a diverse range of proteins including the claudin-family member Kune-Kune (Kune), the FERM-domain protein Yurt (Yrt)[6,7], and the MAGUK Discs Large (Dlg)[8]. As with the aECM, formation and maintenance of SJs require endocytic trafficking[9]. Apically-localized regulators of DT dimensions also interact with the endocytic pathway, including the polarity protein Crumbs (Crb)[7,10–13] and the transmembrane protein Uninflatable (Uif)[14]. The endocytic system therefore plays a central role in regulating diverse tracheal size determinants, but how the endocytic pathway is itself regulated in this context is poorly understood.

One candidate pathway that could regulate intracellular trafficking is the highly conserved Hippo Network (HN), which controls growth in diverse organisms and tissues[15]. Organ growth is promoted upon nuclear translocation of the HN effector Yorkie (Yki) in *Drosophila*, or its mammalian homolog Yes-associated protein (YAP). Yki and YAP are transcription factors that activate genes required for growth and resistance to apoptosis, including the Inhibitors of Apoptosis (IAPs), which inactivate caspases[16]. When Yki/YAP activity is low, organ size is typically reduced due to apoptosis resulting from derepressed caspase activity[17]. However, we previously observed that loss of Yki or Death-associated inhibitor of Apoptosis (Diap1) causes overelongated trachea despite a normal number of cells. Thus, Yki and Diap1 regulate tube size independently of apoptosis[18].

Here, we show that the Caspase-3 homolog DrICE acts downstream of Yki and Diap1 to regulate tracheal elongation. Instead of causing cell death, DrICE regulates endocytic trafficking of tracheal size determinants. This work reveals an intersection of the HN, caspases, and the endocytic system that has critical functions during normal development. Consistent with previous evidence that mammalian caspases can control endocytic trafficking during apoptosis[19,20], we propose a model in which regulation of endocytic trafficking is an evolutionarily conserved function of caspases that can be activated with or without triggering cell death, to contribute to morphogenesis or apoptosis, depending on cellular context.

## Results

**DrICE governs tracheal size downstream of the Hippo Network**. We previously showed that DrICE is required for tracheal dorsal trunk elongation, potentially downstream of the HN[18]. Tracheae fail to elongate normally in embryos homozygous for the *DrICE^{Δ1}* allele, which deletes the DrICE coding sequence[21], but embryos mutant for Yki and Diap1, both of which negatively regulate DrICE, have overly elongated tracheae at stage 16. These results were consistent with, but did not show, that DrICE acts downstream of Yki and Diap1 in tracheal elongation.

We tested whether reduction of DrICE could suppress the long tracheal phenotypes caused by the *yorkie^{B5}* mutation[22], or that of its transcriptional target Diap1, which is encoded by the *thread (th)* locus[18,23,24]. Single loss-of-function *yki^{B5}* or *th^{J5C8}* mutants each have elongated tracheae that follow irregular sinusoidal paths (Fig. 1b–n), but double mutant trachea of the genotypes *yki^{B5}; DrICE17* or *th^{J5C8}, DrICE^{17}* are straight and have either WT lengths or have reduced lengths of *DrICE^{17}* mutants (Fig. 1d–n). These results indicate that DrICE acts downstream of, or in parallel to, Yki, Diap1, and the HN.

For the above experiments, and most of the subsequent experiments in this report, we used the *DrICE^{17}* allele, which has a point mutation[25] in a region near the substrate binding site of DrICE. However, in contrast to a previous report that *DrICE^{17}* is a protein-null allele[25], using different antibodies we find that *DrICE^{17}* generates a stable protein (Fig. 1o and Fig. 2e, f), and causes a stronger tracheal phenotype than *DrICE^{Δ1}*, the null allele[18]. Thus, *DrICE^{17}* behaves as a dominant negative allele that competes with maternally contributed DrICE.

To distinguish between downstream or parallel action of DrICE with respect to the HN, we used western blotting on stage 16 embryos to show that wild-type (WT) DrICE protein levels are elevated by ~50% in both *yki^{B5}* and *th^{J5C8}* mutant embryos (Fig. 1o). Together, these results confirm that DrICE is necessary for dorsal trunk elongation and that DrICE acts downstream of the HN.

**DrICE is sufficient to elongate trachea**. We then asked if DrICE expression is sufficient to drive tracheal elongation without increased upstream HN activity by expressing DrICE in the tracheal system using the UAS-Gal4 system[26], specifically with the *breathless (btl)-Gal4* driver[26,27]. This overexpression results in elongated tracheae, similar to *yki^{B5}* mutants (Figs 1c–n and 2i–j). DrICE is therefore necessary and sufficient to drive tracheal elongation.

**Tracheal cells are not refractory to apoptosis**. Although the executioner caspase DrICE is necessary for tracheal elongation, few (< 3%) tracheal cells undergo apoptosis during wild-type morphogenesis.[28] This raises the question of how a caspase can act in a developmental process without triggering apoptosis. We investigated the possibility that apoptosis maybe disabled in the developing trachea by analyzing loss-of-function mutants for the caspase inhibitor Diap1 (*th^{J5C8}*). We previously showed that *th^{J5C8}* mutant embryos with contiguous dorsal trunks have elongated tracheae despite a WT number of cells[18], but a more comprehensive analysis of *th^{J5C8}* mutants here revealed that the majority of *th^{J5C8}* embryos, are in fact missing dorsal trunk segments (Fig. 1i). Strikingly, 25% of heterozygous[29] *th^{J5C8}* embryos displayed missing tracheal segments (Fig. 1l), suggesting that tracheal cells are capable of undergoing caspase-mediated apoptosis.

We confirmed the existence of apoptosis in *th* mutant tracheae by counting the number of tracheal nuclei. In stage 16 *th^{J5C8}* DT segments, tracheal cell number is reduced by an average of 55% compared to WT (*p* < 0.0001 Student's *t*-test for DT segments 5–6, Fig. 1m). Notably, average DT cell number drops from 26 in WT to 15 in *th* DT segments that are intact, and to only 10 in broken DTs, suggesting that there is a minimal number of cells that is required in order to assemble or maintain DT segments (Fig. 1m). Further evidence that the missing tracheal cells in *th* mutants underwent caspase-mediated apoptosis is provided by the presence of cells with strong cleaved caspase staining in *th* mutants (Supplementary Fig. 2a), and by the 150% suppression of DT breaks and 140% increase in the number of tracheal cells in *th^{J5C8} DrICE^{17}* mutants (Fig. 1m). Thus, the apoptotic roles of Diap1 and DrICE are maintained in tracheal cells during normal morphogenesis.

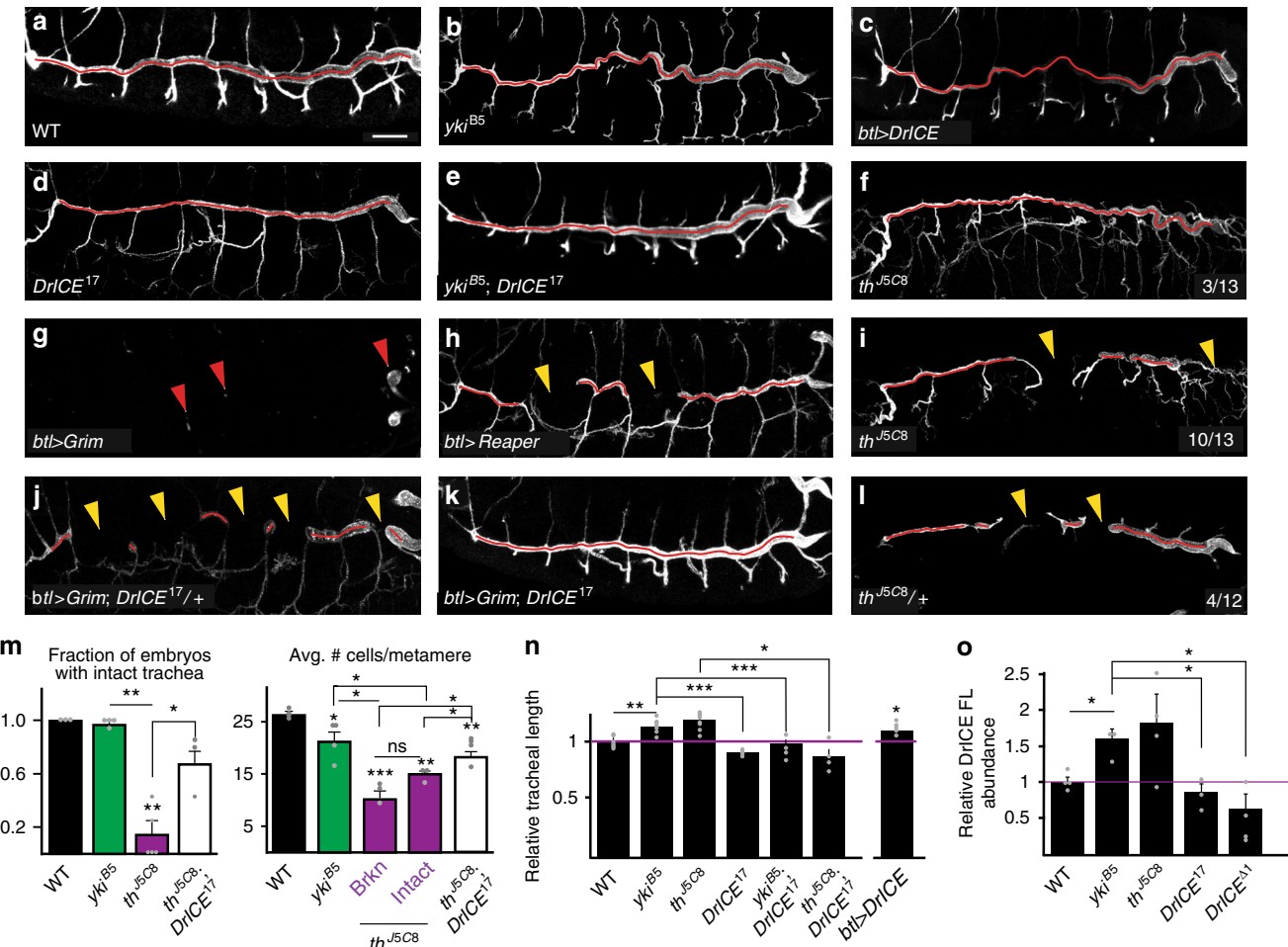

**Fig. 1** DrICE governs tracheal size downstream of Yorkie without triggering apoptosis. **a–l** Compared to WT ($w^{1118}$) (**a**), the dorsal trunks (DTs) of $yki^{B5}$ mutant embryos (**b**), and those overexpressing DrICE in the tracheal system ($btl > DrICE$) (**c**) are elongated, while DTs in $DrICE^{17}$ are too short (**d**). $DrICE^{17}$ is epistatic to $yki^{B5}$ since $yki^{B5}$; $DrICE^{17}$ double mutants do not have long trachea (**e**). Loss of the DrICE inhibitor Diap1/*th* can cause tracheal overelongation (**f**) or missing DT segments when homozygous (**i**) or heterozygous (**l**). Expression of the pro-apoptotic genes $btl > Grim$ (**g**) or $btl > Reaper$ (**h**) causes segment loss dependent on $DrICE^{17}$ dosage (**j**, **k**). Red arrows in **g** mark remnants of the tracheal system, while yellow arrowheads in **h–l** mark missing dorsal trunk segments. Scale bar for **a–l** in **a**, 25 µm. **m** Loss of Diap1 ($th^{J5C8}$) decreases the number of embryos with intact dorsal trunks ($N = 17$ embryos over five experiments) and the number of tracheal cells ($N = 6$ embryos total: 3 broken, 3 intact), indicating apoptosis. The loss of intact trachea and overall cell numbers can be suppressed by $DrICE^{17}$ ($N = 20$ over three experiments and $N = 4$, respectively). Neither cell death nor dorsal trunk breaks are present in $yki^{B5}$ embryos ($N = 4$). **n** DT length in single and double mutant combinations of HN and DrICE mutants. DT length normalized to WT ($w^{1118}$) ($N = 10$) for HN and DrICE mutants, and to $btl$-Gal4, UAS-GFP/+ for overexpressed DrICE ($btl$-Gal4 UAS-GFP/+; UAS-DrICE/+) ($N \geq 5$ for all except $N = 4$ for $th^{J5C8};DrICE^{17}$). **o** Western blot of stage 16 embryos using the α-DrICE$^{CST13085}$ antibody, which recognizes full-length DrICE. A representative blot is shown in Supplementary Fig. 1i. The 47 kDa DrICE full-length band was quantified relative to total protein from at least three experiments. DrICE protein levels in $DrICE^{17}$ homozygotes are not different than WT, consistent with $DrICE^{17}$ being a dominant negative allele that causes more severe tracheal phenotypes than $DrICE^{\Delta118}$ (**n**, **o**). ($N = 4$ blots). All source data are provided in the source data file. For all graphs, each data point is indicated as a point. Error bars, S.E.M. *$p < 0.05$; **$p < 0.005$; ***$p < 0.0005$ Student's $t$-test

Interestingly, in contrast to $th^{J5C8}$ mutants, loss of Yki, which is a key co-activator of Diap1 transcription[30,31], does not induce DT breaks and only reduces tracheal cell number from 26 to 21 (Fig. 1m). This observation suggests that a basal level of Diap1 expression may protect tracheal cells from DrICE-mediated apoptosis in $yki^{B5}$ mutants, yet still allow elevated DrICE activity to elongate tracheal cells.

To further test the apoptotic potential of tracheal cells, we expressed the pro-apoptotic genes Grim and Reaper[32] in the developing trachea using the $btl:Gal4$ tracheal driver. Overexpression of Grim largely eliminates the tracheal system (Fig. 1g). Reaper overexpression results in the loss of one or two tracheal dorsal trunk segments in most embryos, similar to the homozygous $th^{J5C8}$ mutant phenotype (Fig. 1h, i). Critically, the

lethal effects of Grim overexpression are dominantly suppressed by one copy of the $DrICE^{17}$ allele (Fig. 1j), and homozygous $DrICE^{17}$ blocks the ability of Grim to destroy the tracheal system (Fig. 1k). Taken together, these results demonstrate that tracheal cells are not refractory to apoptosis—they have a functional caspase-dependent apoptotic system that does not kill cells during normal development, despite the requirement for DrICE in tracheal elongation.

**Punctate DrICE is enriched at the tracheal apical surface.** To explore the mechanism by which DrICE mediates tracheal elongation, we considered whether DrICE might be spatially compartmentalized in non-apoptotic cells, as is the case for the initiator caspase Dronc, which Amcheslavsky et al. showed was

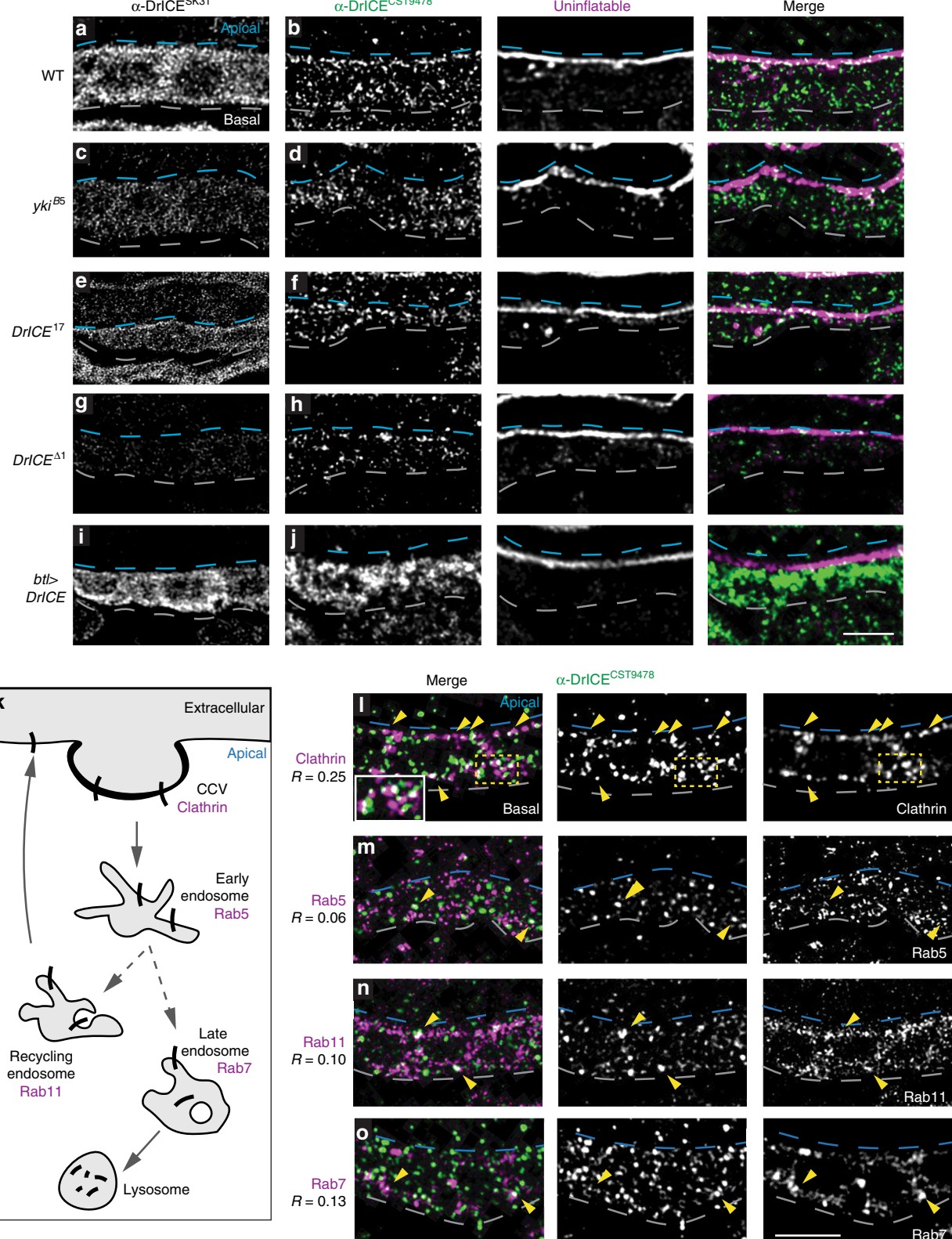

localized basally for non-apoptotic roles in larval imaginal discs[33]. Staining with the α-DrICE[SK31] antibody, raised against full-length DrICE protein[34,35], produces a diffuse cytoplasmic signal in WT trachea that is greatly reduced in the nucleus (Fig. 2a). A different pattern of staining is observed with the α-DrICE[CST9478] antibody, which is raised against a peptide N-terminal to the Asp230 cleavage site of DrICE and is reported to preferentially recognize

the cleaved form of DrICE (Cell Signaling Technologies, #9478). α-DrICE[CST9478] signal is localized to numerous cytosolic punctae that are enriched at the apical surface of tracheal cells, particularly during stages 13 and 14 (Fig. 2b, Supplementary Fig. 1b–k). Thus, there is a pool of DrICE that is localized separately from the inhibitor Diap1, which could potentially allow DrICE to elongate the trachea without triggering apoptosis.

**Fig. 2** A pool of DrICE partially co-localizes with Clathrin. **a–j** Staining with the α-DrICE[SK31] antibody against full-length DrICE (**a–i**) reveals broad cytoplasmic signal in WT trachea (**a**) that is reduced in trachea zygotically homozygous for the null allele DrICE[Δ1] (**g**) and elevated in trachea overexpressing DrICE (**i**). Staining with α-DrICE[CST9478] (**b–j**), which was raised against a peptide containing the DrICE sequence that is cleaved during activation, reveals a more restricted punctate pattern that is enriched at the tracheal apical surface in WT trachea (dashed blue lines), and particularly at earlier developmental stages (Supplemental Fig. 1b, e). α-DrICE[CST9478] signal is reduced in DrICE[Δ1] trachea (**h**), elevated in trachea overexpressing DrICE (**j**). DrICE[CST9478] signal in h results from the maternal contribution of DrICE that is also observed by western blot in Fig. 1o (representative gel image in Supplementary Fig. 1l) and is absent in late larval tissue (Supplementary Fig. 2d–g). **k** Partial schematic of the endocytic system showing the roles of endocytic markers. **l–o** Localization of DrICE[CST9478] signal with markers of the endocytic system in WT (w[1118]) trachea. α-DrICE[CST9478] signal shows partial colocalization with Clathrin (**l**) but not and Rab5 (**m**), with Rab11 (**n**) or Rab7 (**o**). Co-localization was quantified in two ROIs in each of $N \geq 3$ z-slices for $N \geq 4$ embryos. Source data are provided as a source data file. Blue dashed lines mark the apical cell surface, white dashed lines mark the basal cell surface. Scale bar for **a–j** in **j**, 5 μm; for **l–o** in **o**, 5 μm

**DrICE partially co-localizes with Clathrin**. The punctate pattern of α-DrICE[CST9478] signal resembles that of intracellular trafficking machinery, suggesting that this pool of DrICE may associate with trafficking compartments. We tested for colocalization of α-DrICE[CST9478] signal with markers of endocytic vesicle formation (Clathrin), and for early (Rab5), recycling (Rab11), or late (Rab7) endosomes. Punctate DrICE[CST9478] signal partially overlaps with Clathrin both at the apical surface and in cytosolic regions (yellow arrowheads, yellow box and inset in Fig. 2l, respectively; 23% overlap; Pearson $R = 0.25$). However, there is only sporadic co-localization with Rab5 (Fig. 2m; 3.8% overlap, $R = 0.06$), Rab11 (Fig. 2n; 6% overlap, $R = 0.1$), or Rab7 (Fig. 2o; 2% overlap, $R = 0.12$) early, late and recycling endosomes, respectively. The pool of DrICE defined by α-DrICE[CST9478] is therefore positioned to regulate tracheal size by influencing Clathrin-mediated endocytosis and subsequent trafficking of proteins that determine tracheal size. In addition, the incomplete co-localization of DrICE[CST9478] signal with Clathrin suggests that multiple pools of DrICE exist, which supports the idea that DrICE acts in multiple independent functions, including apoptosis.

In addition to its role in early endocytosis, Clathrin also mediates protein trafficking from the transgolgi network (TGN). Because the overlap of Clathrin with DrICE appears to be both apical and cytosolic (Fig. 2l, arrowheads), the intracellular co-localization of α-DrICE[CST9478] signal and Clathrin suggests that DrICE could also influence Clathrin-mediated TGN trafficking.

**DrICE alters the trafficking of apical Crbs and Uif**. Because transmembrane determinants of tracheal tube size including Crb and Uif are known to be trafficked through endocytic compartments[10,11], we asked if DrICE associates with compartments containing these proteins. In WT stage 16 tracheae, DrICE[CST9478] signal partially overlaps with Uif (Fig. 2b, 12% colocalization, Pearson correlation $R = 0.21$; Supplementary Fig. 1h, green) and Crb (Supplementary Fig. 1h, magenta; 17% colocalization, $R = 0.27$), but shows stronger overlap with Crb at the apical surface during tube expansion at stages 13–14 (Supplementary Fig. 1b, d). An apical pool of DrICE is therefore positioned to directly regulate trafficking of apical markers.

We tested if there were functional interactions between DrICE and transmembrane apical markers by examining Crb and Uif in DrICE[17] mutants. Overall apical/basal polarity as assessed by Crb staining is largely unaffected in DrICE[17] mutations. Levels of apical and cytosolic Crb are not significantly altered in DrICE[17] mutants at stage 16, but in stage 14 embryos, fewer Crb-positive cytosolic punctae were visible in DrICE[17] mutants than in WT ($p < 0.05$ Student's t-test, Fig. 3a–j) and conversely more punctae were present in yki[B5] mutant trachea than in WT ($p < 0.005$ Student's t-test, Fig. 3a–j). Similarly, overexpression of DrICE in the tracheal system increases Crb levels at stage 16 ($p < 0.0005$ Student's t-test, Fig. 3l–r). No significant changes in Uif

abundance or distribution were observed in DrICE mutant tracheae (Fig. 2f, h), but DrICE[17] suppresses the increased abundance of Uif positive punctae in the tracheae of embryos mutant for the basolateral polarity protein Yurt ($p < 0.005$ Student's t-test; Fig. 4a–i). Moreover, DrICE[17] also suppresses the increased tracheal length of the yrt[65A] mutants ($p < 0.005$ Student's t-test, Fig. 4j).

**DrICE alters the trafficking of the claudin Kune**. We tested whether DrICE mutants exhibited alterations in trafficking of basolateral junctional components, which are known to be constitutively recycled through the endocytic system through tracheal development[36,37]. Immunostaining for the claudin Kune-Kune (Kune) revealed that junctional and overall levels of Kune are reduced in yki[B5] trachea ($p < 0.05$ Student's t-test; Fig. 3e–k), and in trachea overexpressing DrICE (Fig. 3p), both conditions in which DrICE is elevated. Conversely, in DrICE[17] mutants, strong ectopic staining for Kune is apparent at the apical (lumenal) surface and overall levels of Kune are increased (Fig. 3f–k). However, not all SJ components are negatively regulated by DrICE. For example, levels of the MAGUK Discs Large (Dlg) were only modestly affected in yki[B5] and DrICE[17] mutants (Fig. 3g–i), and we previously reported that the FERM-domain protein Coracle was unaffected by Yki and DrICE mutations. Thus, DrICE does not globally regulate endocytic trafficking, but rather is required for trafficking of a subset of tracheal markers.

**DrICE alters the trafficking of Serpentine**. We investigated whether DrICE is required for trafficking of lumenal determinants required for tracheal size control, including the putative chitin deacetylase Serpentine (Serp). As with the apical determinant Uif, we found that DrICE[17] suppresses the overabundance of Serp-containing punctae in yrt[65A] ($p < 0.05$ Student's t-test; Fig. 4e–i). Furthermore, overactivation of DrICE resulting from loss of yki decreases the amount of lumenal Serp in tracheal anterior, a phenotype that is partially suppressed by DrICE[17] ($p < 0.005$ Student's t-test; Fig. 4k and green in Fig. 4l–o). DrICE therefore affects trafficking of lumenal tracheal size determinants and mediates the effects of the HN on their trafficking.

Together, these results reveal that DrICE activity selectively modulates the trafficking of cargo originating from both the apical and basolateral membranes, and at minimum affects the trafficking of at least four different proteins required for tracheal size control.

## Discussion

Our work shows caspase-mediated regulation of endocytic trafficking during normal morphogenesis. Moreover, the caspase activity responsible for the endocytic function does not trigger apoptosis, even though the cells are competent to undergo caspase-mediated apoptosis. We also demonstrate a link between

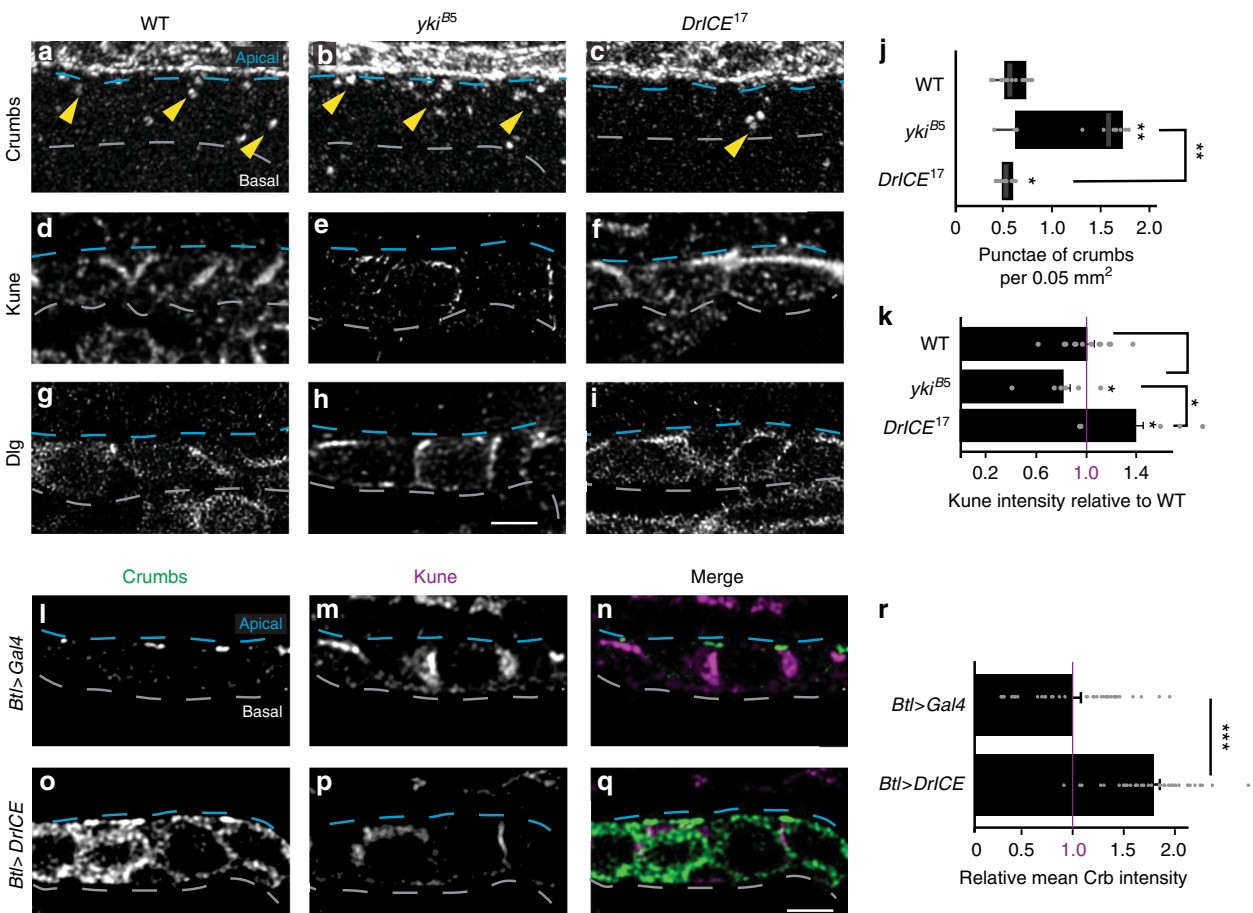

**Fig. 3** DrICE regulates abundance and localization of the size determinants Crb and Kune. **a–c** Number of Crb cytosolic punctae (yellow arrowheads) is increased in $yki^{B5}$ mutants (**b**) but decreased in $DrICE^{17}$ mutants (**c**) compared to WT ($w^{1118}$) (**a**) at stage 14. **d–f** Cytoplasmic Kune abundance is decreased in $yki^{B5}$ and increased in $DrICE^{17}$ relative to $w^{1118}$. Quantification in **k**. **g–i** Levels of the basolateral septate junction protein Dlg appear minimally affected in $yki^{B5}$ and $DrICE^{17}$ mutants, indicating that the ability of DrICE to regulate trafficking is selective. **j** The number of cytoplasmic punctae of Crumbs at embryonic stage 14 is reduced in $DrICE^{17}$ tracheal cells compared to $w^{1118}$ and increased in $yki^{B5}$ relative to $w^{1118}$. Punctae were counted in two ROIs in each of two z-slices for $N \geq 5$ embryos and normalized to ROI area. Center line represents the median, box limits are the upper and lower quartiles, whiskers are 1.5× interquartile range. **k** Mean intensity of Kune in a maximum projection of the fifth tracheal metamere was measured in $N \geq 5$ embryos of each genotype then normalized to the WT ($w^{1118}$) control. **l–q** Overexpression of DrICE in the trachea increases Crb levels relative to the control at stage 16 (**l** vs. **o**), consistent with the decrease of Crb punctae in stage 14 $DrICE^{17}$ mutants. Quantified in **r**. Conversely, Kune levels are decreased when DrICE is overexpressed (**m** vs. **p**). **r** Cytoplasmic Crb abundance is dramatically increased in btl>DrICE relative to btl>Gal4/+. Mean Kune intensity was measured in two ROIs in each of five z-slices for $N = 4$ embryos of each genotype and quantified relative to the btl>Gal4/+ control. For all graphs, error bars represent S.E.M. and each data point is represented as a point. Source data are provided as a source data file. Blues dashed lines mark the apical cell surface, white dashed lines mark the basal cell surface. Scale bar for **a–i** in **i**, 2.5 μm; for **l–m** in **m**, 2.5 μm. *$p < 0.05$; **$p < 0.005$; ***$p < 0.0005$ Student's t-test

the HN and the endocytic system. While previous studies focused on the effects of the endocytic system on the output of the Hippo Network[38–41], we report the converse: the Hippo Network acts as input to the endocytic system that alters endocytic trafficking.

A notable aspect of the role of DrICE in trafficking is that DrICE is required for trafficking of select—not all—trafficked cargo. For example, the abundance and localization of the SJ protein Kune is affected by alterations in DrICE, but Coracle and Dlg are not strongly affected (Fig. 3i, j and see ref. [18]). We also observe defects in the trafficking of the lumenal protein Serp, but no defects have been observed in the trafficking of the similar chitin deacetylase Vermiform (Verm)[18]. The differential trafficking of Serp and Verm in DrICE mutants is commensurate with observations by Dong et al.[11], who observed that a mutation in shrub/VPS32 that disrupts multivesicular body formation also altered the localization of Serp, but not Verm. This similar differential effect on Verm and Serp trafficking provides further evidence of the involvement of DrICE in endocytosis. It is also

notable that not only does DrICE regulate specific cargos, the effect on each cargo is also cargo specific. For example, DrICE activity promotes Crb accumulation, but suppresses Kune accumulation. These results are consistent with DrICE acting directly on individual cargos, cargo-adaptors such as sorting adaptors, or on both cargo and adaptors.

The counterintuitive requirement of an executioner caspase to promote growth in an epithelium has not previously been reported in mammals. However, there is in vitro evidence that mammalian caspases regulate endocytic trafficking[20]. For example, Duclos et al. showed that activation of activator and effector caspases in HeLa and HEK293 cells resulted in cleavage of Sorting Nexins 1 and 2 (SNX1 and SNX2), which control intracellular trafficking of specific cargos[19]. Cleaved SNX increased activation of the hepatocyte growth factor receptor (HGFR) and ERK1/2 signaling. Additionally, Han et al. showed that Caspase-3 cleavage of junctional component Gap43 was required for endocytosis of the AMPA receptor A in non-apoptotic cells

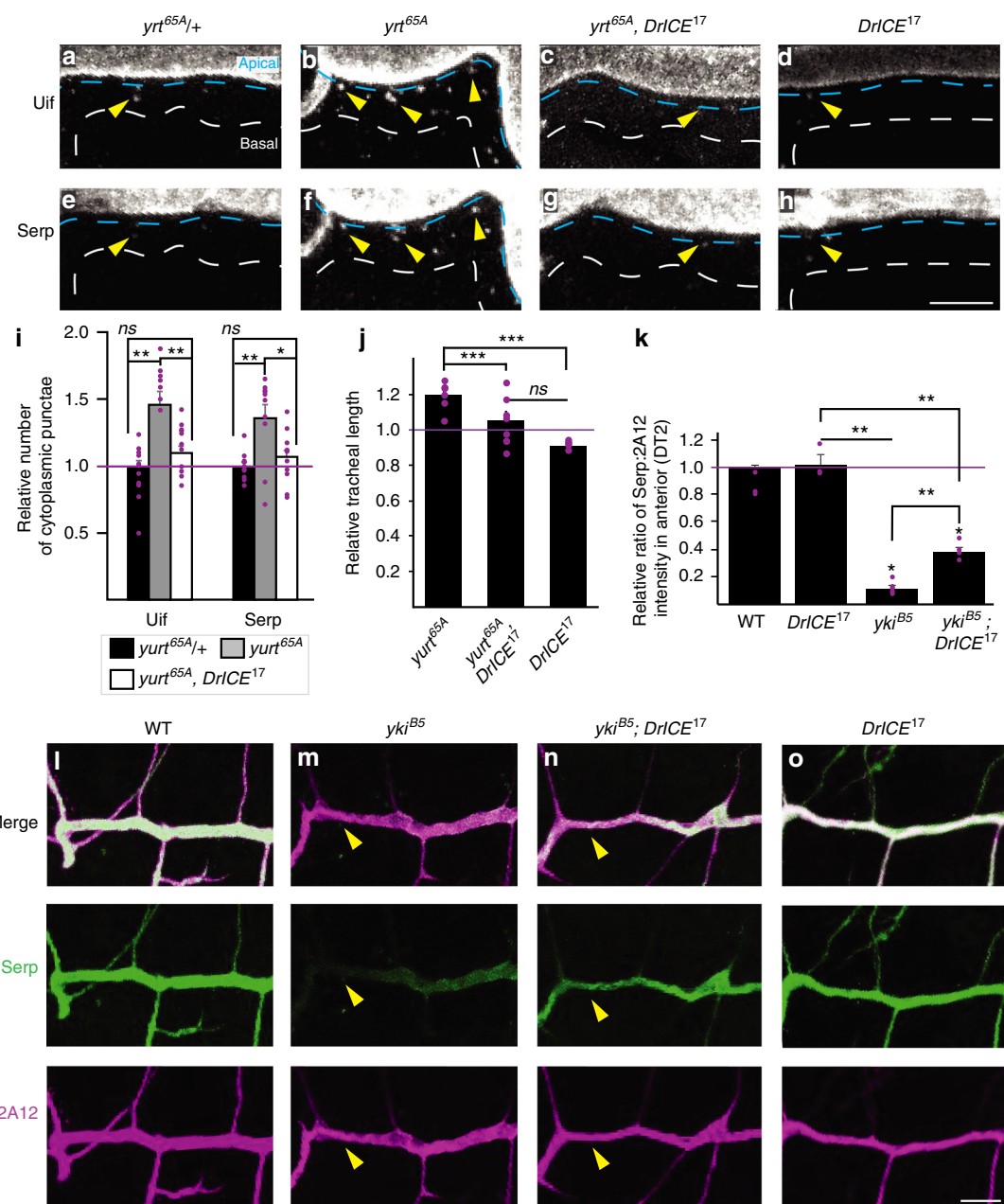

**Fig. 4** DrICE is required for trafficking of tracheal size determinants Serpentine and Uif. **a–h** Mutation of the basolateral polarity protein Yurt (*yrt^65A*) causes abnormal intracellular accumulation of the tracheal size determinants Uif (**b**, arrowhead) and Serp (**f**, arrowhead), an effect that is suppressed by *DrICE^17* (**c, g**). White dashed lines outline basal cell surface; blue dashed lines outline apical surface. Scale bar **a–h** in **h**, 5 μm. **i** Relative number of cytoplasmic punctae quantified, where the number of punctae in each of two ROIs in a maximum projection of *yrt^65A* (*N* = 5 embryos), *yrt^65A*/+ (*N* = 5 embryos), and *yrt^65A*, *DrICE^17* (*N* = 6 embryos) was counted, then divided by the area of the given ROI. Resulting number of puncta per area is reported relative to *yrt^65A*/+. **j** The *DrICE^17* mutation suppresses tracheal overelongation caused by a mutation in the basolateral polarity gene *yrt*. Quantification of tracheal DT length at stage 16 for *yurt^65A* (*N* = 6) and *yurt^65A*, *DrICE^17* (*N* = 7) and *DrICE^17* (*N* = 5) mutants reveals that *yurt^65A*, *DrICE^17* double mutants are shorter than *yurt^65A* mutants. Tracheal lengths normalized to body length and reported relative to WT (*w^1118*). For all graphs, each data point is represented as a point. **k–o** Mutation of the Hippo Network component Yki (*yki^B5*) reduces the levels of Serp (green) in the tracheal anterior relative to the lumenal chitin binding protein Gasp, marked by staining with the monoclonal antibody 2A12 (arrowhead, **m**). This reduction is suppressed by *DrICE^17* (**n**). **k** Quantification of the mean intensity of Serp divided by the mean intensity of 2A12 in a maximum projection of *N* = 4 embryos for WT (*w^1118*), *N* = 4 embryos for *yki^B5*; *DrICE^17*, *N* = 7 embryos for *yki^B5*, *N* = 3 embryos for *DrICE^17*; Numbers reported as relative to *w^1118*. Scale bar **l–o** in **o**, 12.5 μm. For all graphs, error bars represent S. E.M. and each data point is represented as a point. Source data are provided as a source data file. *p* < 0.05; \*\**p* < 0.005; \*\*\**p* < 0.0005 Student's *t*-test

in vitro[42]. Together, the previously published in vitro mammalian results and these in vivo *Drosophila* results suggest a developmental role for caspase-mediated regulation of endocytic trafficking that is evolutionarily conserved and may also function in mammalian development.

The conserved involvement of caspases in endocytic trafficking raises the possibility that there is a deeper mechanistic connection between apoptosis and trafficking of junctional components. In the current model of apoptosis, caspase-mediated proteolysis of cell junctions results in decreased cell adhesion, which allows

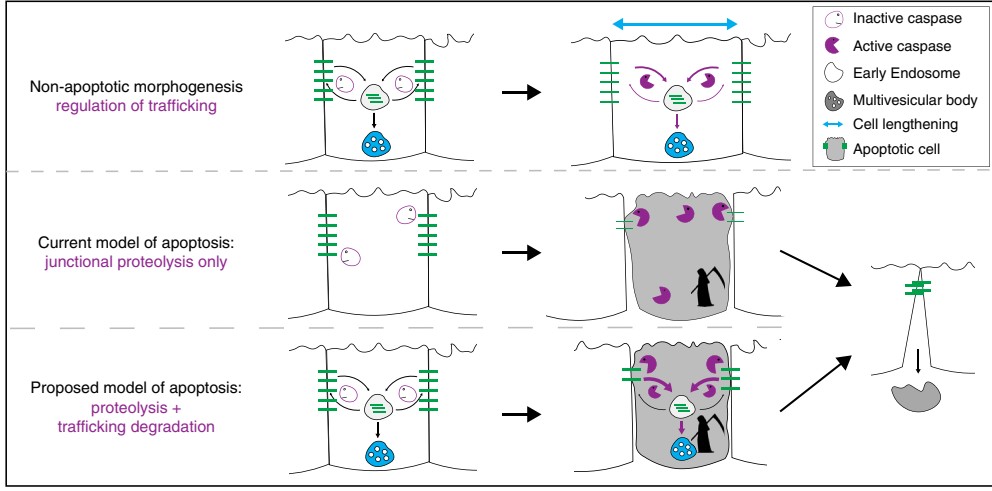

**Fig. 5** Model for caspase-regulated endocytic trafficking in apoptosis. See main text for details

an apoptotic cell to be cleanly extruded from an epithelium (Fig. 5, middle; reviewed by ref. [43]). However, in mammals and flies, junctional components are constitutively cycled through the endocytic system even in non-apoptotic epithelia[36,37,44]. This presents an opportunity for caspases to alter trafficking of junctional components. Perhaps the ability of caspases to regulate endocytic trafficking arose as an ancient apoptotic function that allowed dying epithelial cells to more efficiently downregulate junctions by using both endocytosis and caspase-mediated proteolysis rather than caspase-mediated proteolysis alone (Fig. 5, bottom). We hypothesize that caspase-mediated alteration of cell junction trafficking has been co-opted in morphogenesis, when growing epithelia require rapid remodeling of junctional components (Fig. 5, top).

Experimental evidence from mammalian cerebral ischemic injury supports the proposed model that limited activation of caspases alters junctional endocytosis. During cerebral ischemic injury, blood flow in the brain is blocked, which leads to extensive caspase activation, downregulation of claudin-based tight junctions, and a large and damaging increase in vascular permeability[45]. Pretreatment of tissue with caspase inhibitors increases junctional continuity and integrity including junction components claudin-5 and the tight junction marker ZO-1, and ultimately reduces tissue damage. Strikingly, more endothelial cells show caspase activation than undergo apoptosis, suggesting that caspase activation is sufficient to induce junctional downregulation and vascular permeability.

The ability of caspases to regulate endocytic trafficking has significant implications for cancer treatments, which frequently result in caspase activation. Sub-apoptotic activation of effector caspases in surviving tumor cells after anti-cancer treatment could potentially result in reduced junctional integrity, which could further promote epithelial-to-mesenchymal transformation (EMT) and metastasis. Additionally, caspase-driven changes in the trafficking of polarity proteins such as Crb and of growth factor receptors could also contribute to EMT and disease progression. Consistent with this possibility, previous evidence has indicated that human Caspase-3 is required for migration, invasion and metastasis HCT116 colon cancer cells[46], and for tumor cell repopulation upon irradiation[47].

In summary, this work reveals a function of the effector caspase DrICE during development: modulating endocytic trafficking of cell junctional components and signaling molecules. Given that mammalian effector caspases can also modulate endocytic trafficking, we propose that regulation of the trafficking of junctional

components is a conserved caspase function that can be brought into play during normal morphogenesis or in pathologic conditions by localized, sub-apoptotic caspase activation.

## Methods

**Fly stocks**. The following mutant alleles were used in this paper: $th^{J5C8}$, $yorkie^{B5}$, $DrICE^{17}$, $DrICE^{\Delta1}$, and $yurt^{65A}$. DrICE overexpression in the trachea was achieved using the UAS/Gal4 system: the $btl > Gal4$ driver was used to express Gal4 in all tracheal cells; $btl>Gal4$ flies were crossed to $UAS$-$DrICE$ flies, which had a transgenic DrICE cDNA downstream of UAS (see the section 'Transgenic constructs' below). The double mutant $yurt^{65A}$, $DrICE^{17}$ was generated using standard recombination crosses. Each single male resulting from the recombination crosses was screened for $yurt^{65A}$ via failure to a complement $yurt^{65A}$ single mutant and screened for the presence of the $DrICE^{17}$ mutation by sequencing to identify the point mutation N116Y.

**Transgenic constructs**. To create the UAS-DrICE transgenic line, full-length DrICE cDNA sequence was cloned into the MCS of pUAST vector using Gibson cloning and subsequently inserted into the attP2 site into flies with the phiC31 integrase system.

**Embryo staining**. Parental flies were placed in empty food bottles capped with a molasses-agar plate spread with wet yeast. After 24 h, plates were removed and embryos were fixed and stained as follows: embryos were dechorionated with 50% bleach for 4 min, and fixed in equal parts heptane and 4% formaldehyde on a shaker for 25 min. Embryos were then devitellenized for 1 min with 100% methanol/heptane, rehydrated to PBS-T (0.1% Triton-X100) (100%, 100%, 100%, 50%, each for 5 min), washed in PBST for 5 × 5′ and then 3 × 30′. PBS-BT (3% BSA in PBS-T) was used to block for 30 min at RT before addition of primary antibody, in which embryos were incubated 48 h at 4 °C on a rotator. See Supplementary Table 1 for appropriate antibody dilutions. PBS-T and PBS-BT steps were repeated on the third day before addition of fluorescent secondary antibodies, in which embryos were incubated O/N at 4 °C. The use of the Alexa + secondary antibodies (see chart for details) was essential to visualize DrICE α-DrICE$^{CST9478}$ staining as regular Alexa secondary antibodies did not produce adequate signal-to-noise. On the final day, PBS-T washes were repeated, embryos were dehydrated using an ethanol series (50%, 70%, 90%, 100%, 100% EtOH, each for 5 min), incubated in methyl salicylate for 15 min at RT, and then mounted with methyl salicylate and Permount mounting media (Fisher Scientific, SP15-500). More information about reagents can be found in Supplementary Table 1.

**Image acquisition**. Confocal z-stacks were obtained with the ×63 NA 1.24 oil objective on the Leica SP8 image format ≥2048 × 2048, pinhole size 0.7 AU, and system optimized z-step sizes of ~0.2 µm. Images were subsequently deconvolved using Leica HyVolution software with a Huygens Essential Automatic setting. All quantification was performed on the resulting images.

**Image quantification and general statistics**. Quantification of junctional components was performed using ImageJ, in which a region of interest in the fifth dorsal trunk metamere of tracheal cells was used to calculate the mean pixel intensity of each junctional component. The number of punctae for Crumbs, Uif, or Serp was calculated by acquiring the number of spots with the ImageJ

SpotCounter plugin (Gaussian pre-filtering, box size = 8, and noise tolerance = 20) and then dividing that number by the total area of that region of interest. Each individual embryo's ratio of spots per unit area was then divided by the WT average, and a Student's two-tailed *t*-test was used to calculate the *p*-value.

Tracheal lengths were determined by tracing the length of the dorsal trunk through 3D in Volocity Demo 5.5.1.

**Western blot.** Parental flies (heterozygotes with the CyO GMR-dfd YFP or TM6B GMR-dfd YFP balancers were placed in empty food bottles capped with a molasses-agar plate spread with wet yeast. After 1 h, plates were removed and stored in a humid chamber at 25 °C for 18 h. Embryos were then washed with water and treated with 50% bleach for 4 min. The bleached embryos were sorted to collect only homozygous embryos, which lacked the fluorescent marker denoting the presence of the balancer chromosome. Approximately 30 embryos of each genotype were lysed in RIPA buffer using a micropestle, and then incubated on ice for 30 min before 15-min centrifugation at 4 °C. Protein concentration of the resulting supernatant was measured by Nanodrop 2000 Spectrophotometer (ThermoFisher Scientific), and all samples were diluted with RIPA buffer to the same protein concentration before Laemmeli Buffer was added. Samples were denatured using in a boiling water bath for 5 min, loaded in a gradient gel and run for 90 min at 100 V. Transfer of proteins to the nitrocellulose membrane was carried out at 60 V for 20 min at 4 °C in order to retain low molecular weight caspases. After total protein levels were detected and imaged using the LiCor REVERT total protein stain and Odyssey CLx imaging system (Li-Cor), blots were incubated in blocking solution (5% BSA in TBS with 0.1% Tween20) for 1 h, and then in rabbit anti-*Drosophila* DrICE[CST13085] (1:1000) overnight at 4 °C. Fluorescent Li-Cor goat anti-Rabbit IRDye CW800 secondary antibody was then used to detect the resulting bands, and the mean signal for the 47 kd band of each genotype was then normalized to the corresponding lane's total protein signal. More information about reagents can be found in Supplementary Table 1.

The work has complied with all relevant ethical regulations for animal research. Because *D. melanogaster* are an invertebrate species, no institutional approval of experimental protocols was required.

**Reporting summary**. Further information on experimental design is available in the Nature Research Reporting Summary linked to this article.

## Data availability

The authors declare that the main data supporting the findings of this study are available within the article and its Supplementary Information files. Extra data are available from the corresponding author upon request[48].

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

## Acknowledgements

For reagents we thank Rob Ward (α-Uif), Pascal Meier (α-DrICE[SK31]), Christos Sama-kovlis (α-MTf), and Matthias Behr (α-Clathrin). We also thank Benjamin Kraft for assistance in epistasis experiments; Andreas Bergmann for helpful discussions; J. Hornick and the Northwestern University Biological Imaging Facility for technical and imaging support; and Laura Lackner, Heike Fölsch, and I. Tanelli Helenius for comments on the manuscript. Stocks obtained from the Bloomington Drosophila Stock Center (NIH P40OD018537) were used in this study, and FlyBase was a critical resource. This work was supported by NIH RO1GM108964 to G.J.B. S.S.M. was supported by the National Institute of General Medical Sciences training grant T32GM008061.

## Author contributions

S.S.M. and G.J.B. conceived and designed the experiments. S.S.M. performed the experiments. S.S.M. and G.J.B. analyzed the data. S.S.M. and G.J.B. wrote the manuscript.

## Additional information

**Competing interests:** The authors declare no competing interests.

