## [Peer Review File · Nature Communications]

Reviewers' comments:

Reviewer #1 (Remarks to the Author):

The manuscript by McSharry and Beitel follows up on previous work from regarding the role of the Hippo pathway and caspases on tracheal tube size in the *Drosophila* embryo. The authors reveal a non-apoptotic role of effector caspase DrICE downstream of the Hippo pathway in the regulation of endocytic trafficking, which affects levels and localization of cell polarity and matrix components that are involved in size control. They propose a very interesting hypothesis: that the regulation of endocytic trafficking by caspases is an evolutionary conserved role of caspases that can function during apoptosis (promoting cell extrusion), in non-apoptotic scenarios (regulating morphogenesis in the trachea) and in disease (potentially promoting metastasis). Overall this is a solid and interesting paper that well suited for publication in Nature Communications. However, a few points should be addressed prior to publication:

- While it seems clear that tracheal cells can undergo apoptosis when pro-apoptotic genes are overexpressed, it is less clear whether DIAP1 and DrICE have an apoptotic role under other conditions. Are the missing segments in th mutant embryos a consequence of apoptosis? Does this happen in Yki mutant embryos as well? Is caspase-3 staining or other apoptotic marker seen in these segments before they disappear? Can this phenotype be rescued in a DrICE mutant background?
- The complementary localization in the nucleus vs. cytoplasm of DIAP1 and DrICE would certainly suggest a possible mechanism to explain a non-apoptotic role of effector caspases. However, Fig 2o is not very convincing to support this statement. While some of the cleaved-caspase3-containing puncta don't co-localize with DIAP1, it seems that most of them do. Maybe a better image or a co-staining with DAPI or other nuclear marker could clarify this point.
- It's interesting that these effects are not restricted to the tracheal system: the epidermis and the hindgut show similar defects in endocytic trafficking. But is there any effect on size regulation on these tissues? Or any other organ phenotype of DrICE and/or Yki mutants?
- The basal side of the plasma membrane has been described as a compartment where caspases are localized for non-apoptotic roles (Amchelavsky et al., 2018). Do the authors observe this pattern of localization in the trachea at any stage?
- This current report seems to be in conflict with a previous study on the involvement of SJ and apico-basal polarity pathways in the Yki phenotypes (Robbins et al., 2014). The current work looks more convincing to me, but it would be nice to repeat some of the previous experiments to see if some phenotypes could have been previously missed. At the very least, this issue should be discussed and any clarifications regarding these apparently conflicting results would be helpful to include in the Discussion.

Reviewer #2 (Remarks to the Author):

McSharry and Beitel provide strong genetic evidence for caspase activity that promotes endocytosis in the *Drosophila* tracheal system. The tracheal system is shown to be capable of apoptosis, but normally the caspase DrICE plays an alternate role. The role in endocytosis is evident from altered distributions of Crb in the trachea and epidermis of DrICE mutants, and with DrICE over-expression. DrICE loss of function also suppresses abnormal endocytosis that otherwise occurs in yurt mutants. Altered distributions of further markers are also shown in DrICE mutants. Full-length DrICE displays a diffuse cytosolic distribution, but antibodies for cleaved DrICE reveal a punctate pattern with enrichment to the apical domain and some co-localization with clathrin and Rab5. Yorkie and its transcriptional target Diap1 (an inhibitor of DrICE) both inhibit total levels of DrICE, as revealed by mutant analyses. Their mutants also display contorted dorsal trunks of the tracheal system, a defect that can be suppressed with a DrICE mutant or mimicked with DrICE overexpression but does not seem to be due to apoptosis. Through single and double mutant analyses, DrICE is shown to be responsible for a reduced deposition of luminal material in Yorkie mutants. Overall, a demonstration of a role for a caspase in

endocytosis in a whole animal should be of general interest. However, a number of issues should be addressed.

1. It was unclear how to interpret the comparisons of the aPKC and pSrc distributions in the shrub and DrICE mutants. Shrubby, aPKC and pSrc seem to be indirectly associated with endocytosis. Can the authors clarify their rationales and conclusions in the manuscript?
2. The authors claim that “pSrc is reduced at the apical membrane in both DrICE17 and shrubby mutant trachea (Fig. 3g-i)”, but the data are not convincing.
3. The colocalization of cleaved DrICE with clathrin and Rab5 was unconvincing. It seems that it could be coincidental. Also, there are many puncta of the cleaved DrICE through the cytosol that lack colocalization with any markers of the endosomal system tested. Could a marker for bulk endocytosis reveal more colocalization? This or other analyses are important for consolidating the endocytosis connection. Also, the degree of colocalization is important for the authors' model for a “mechanism by which tracheal cells avoid death in the face of activated caspases...[in which]...DrICE localizes to the endocytic system, which may help limit DrICE function to the endocytic pathway”.
4. There are substantial puncta detected with the antibody for the cleaved DrICE even in a null mutant for DrICE that lacks staining for total DrICE. To what degree do these non-specific puncta colocalize with endocytic markers, or with Crumbs (relevant to Fig S1)?
5. The last section of the results briefly describes analyses in other tissues. The section and the data are difficult to interpret because of the minimal explanations by the authors. Moreover, the investigations of the colocalization between the cleaved DrICE and the endocytic markers in the hindgut (Fig S3) have the issues raised above, but additionally lack any quantification.
6. A typo: in “over-expression of DrICE in the tracheal system increases Crb levels at stage 16 ($p < 0.0005$, Supplementary Fig. 4a-c)”, “Supplementary” should be deleted.

Reviewer #3 (Remarks to the Author):

McSharry and Beitel

This is an interesting manuscript, which follows up on their discovery of a function for Yki and its target gene DIAP1 in restricting tracheal tube elongation by antagonising DrICE, which acts downstream of DIAP1 to promote elongation (PLoS One 2014 9(7): e101609). The new manuscript contains a great deal of data but most importantly reveals a physiological function for DrICE in promoting Crb endocytosis in trachea that is complementary to its normal role in promoting apoptotic cell death in other tissues (Fig 3a and 4a-c). The loss-of-function phenotype of DrICE mutants can be rescued by simultaneous mutation of Yurt, another protein that is known to promote Crb endocytosis (Fig 4h-l). Overall, these are important new findings that should be published in Nature Communications subject to the following major revisions.

Major comments:

1. There are currently 6 highly complex figures that need to be reduced to 4 simple ones. Fig 1 is fine as is. Fig 2 should be placed in Supplementary Figures. Figs 3a and 4a-c should be combined into a new Fig 2 that is simple and emphasises the most significant finding of the manuscript - regulation of Crb by DrICE. The remainder of Fig 3 (3d-i and 3j-r) should be removed from the manuscript. The remainder of Fig 4 (4d-l) should become a simplified new Fig 3 that highlights only the rescue by yurt, leaving out extraneous data on Serp, Kune-Kune and Mtf (which are not in the least convincing). The

current model Fig 6 then becomes the new Fig 4. Current Fig 5 should be removed from the manuscript.

2. The effect of DrICE on aPKC, p-Src, Serp and Uif was not convincing and should simply be removed from the manuscript.

3. The effect of DrICE on Crb outside of the trachea (e.g.: Hindgut Fig S4 or Epidermis Fig 3j,k) seemed much less clear than in the trachea itself. I would suggest either removing this data from the manuscript or making a single separate supplementary figure that clarifies the tracheal-specific nature of this phenotype.

4. The abstract needs to be amended to focus on Crb, and remove reference to 'at least seven ... proteins', as the data only support a clear effect on Crb (which could indirectly affect others).

Reviewer #1

The manuscript by McSharry and Beitel follows up on previous work from regarding the role of the Hippo pathway and caspases on tracheal tube size in the *Drosophila* embryo. The authors reveal a non-apoptotic role of effector caspase DrICE downstream of the Hippo pathway in the regulation of endocytic trafficking, which affects levels and localization of cell polarity and matrix components that are involved in size control. They propose a very interesting hypothesis: that the regulation of endocytic trafficking by caspases is an evolutionary conserved role of caspases that can function during apoptosis (promoting cell extrusion), in non-apoptotic scenarios (regulating morphogenesis in the trachea) and in disease (potentially promoting metastasis). Overall this is a solid and interesting paper that well suited for publication in *Nature Communications*.

We thank the reviewer for their support.

However, a few points should be addressed prior to publication:

- While it seems clear that tracheal cells can undergo apoptosis when pro-apoptotic genes are

overexpressed, it is less clear whether DIAP1 and DrICE have an apoptotic role under other conditions. Are the missing segments in *th* mutant embryos a consequence of apoptosis? Does this happen in *Yki* mutant embryos as well? Is caspase-3 staining or other apoptotic marker seen in these segments before they disappear? Can this phenotype be rescued in a DrICE mutant background?

These are excellent questions. In response, we have characterized tracheal cell apoptosis in *th* and *yki* mutant embryos by counting tracheal nuclei and staining with an anti-cleaved caspase antibody that recognizes cleaved caspases and caspase substrates and has previously been used as a marker for apoptotic cells (new panel 1m, new panels S2a-c). These results show that there is indeed apoptosis of tracheal cells in *th* mutants, and in particular in tracheal segments with broken branches. In contrast, tracheal cell numbers are normal in *yki* embryos, which do not have dorsal trunk breaks. The loss of tracheal cells in *th* is suppressed by a DrICE mutation. Thus, DIAP1 and DrICE have apoptotic roles under more physiological conditions as well as when pro-apoptotic genes are overexpressed.¹ These observations are detailed on pages 6-7 of the revised manuscript.

Are the missing segments in *th* mutant embryos a consequence of apoptosis? Can this phenotype be rescued in a DrICE mutant background?

Yes and yes. Approximately 80% of DIAP (*th*) mutant trachea have broken segments, but the frequency of breaks is reduced to 25% in *th*^{J5C8}, *DrICE*¹⁷ double-mutant embryos (new panel 1m). To demonstrate that the missing segments in *th* mutant embryos are in fact a consequence of apoptosis, we quantified the number of tracheal cells in *th* mutants and found that the number of cells was reduced from 26.4 to an overall average of 12.7 in *th* mutants. Strikingly, broken DT segments in *th* mutants only have an average of 10.3 cells compared to an average of 15 for intact *th* branches (new panel 1m). Removing DrICE from the *th* background increases tracheal cell number in *th*^{J5C8}, *DrICE*¹⁷ double-mutant embryos by 40% compared the *th* mutant embryos alone (new panel 1m). The text has been updated accordingly on pages 6-7.

Is caspase-3 staining or other apoptotic marker seen in these segments before they disappear?

We used the cleaved caspase antibody as a marker of apoptosis¹ to stain for potential apoptotic cells in *th*, *yki* and *DrICE* mutant embryos. We examined several stages of development and found that at any given timepoint, a few more tracheal cells in *th*, but not *yki* or *DrICE* showed high levels of cleaved caspase staining. These results are consistent with tracheal cells activating caspases before undergoing apoptosis and are shown in Fig. S2a-c and discussed on page 6 of the revised manuscript.

- The complementary localization in the nucleus vs. cytoplasm of DIAP1 and DrICE would certainly suggest a possible mechanism to explain a non-apoptotic role of effector caspases. However, Fig 2o is not very convincing to support this statement. While some of the cleaved-caspase3-containing puncta don't co-localize with DIAP1, it seems that most of them do. Maybe a better image or a co-staining with DAPI or other nuclear marker could clarify this point.

Because this not a central point of the paper and we do not have data proving that compartmentalization contributes to the non-apoptotic role(s) of DrICE, we have removed the DIAP1 staining from the figures and the text in order to focus on more critical content.

- It's interesting that these effects are not restricted to the tracheal system: the epidermis and the hindgut show similar defects in endocytic trafficking. But is there any effect on size regulation on these tissues? Or any other organ phenotype of DrICE and/or *Yki* mutants?

There is no obvious effects of Yki or DrICE mutations on size regulation in the epidermis or in the hindgut, but this is consistent with the lack of phenotypes in these tissues in mutants of the markers that we have shown are mistrafficked in DrICE mutants. For example, septate junction mutants, including Kune, do not affect hindgut or epidermal size. Another consideration is that the overall effect of DrICE mutants of trafficking in the hindgut and epidermis is weaker than in the trachea, possible because the executioner caspase DCP-1 is also expressed in these tissues. It may be necessary to eliminate both caspases to cause strong mistrafficking of determinants and thus size phenotypes in non-tracheal tissues². Regardless, while we are also very interested in non-tracheal phenotypes, reviewers 2 and 3 requested that the manuscript be focused on tracheal effects, and we have eliminated the non-tracheal data in the revised manuscript.

- The basal side of the plasma membrane has been described as a compartment where caspases are localized for non-apoptotic roles (Amchelavsky et al., 2018). Do the authors observe this pattern of localization in the trachea at any stage?

This comment addresses the observation by Amcheslavsky et al 2018 that the basal side of the plasma membrane can serve as a compartment where the initiator caspase Dronc is localized for non-apoptotic roles in imaginal discs³. In the trachea, however, we find that the executioner DrICE is not particularly concentrated at the basal side of the plasma membrane. We have added this point on page 7 of the manuscript, and have added dotted lines to outline the apical and basal sides of tracheal cells for clarification in all images.

- This current report seems to be in conflict with a previous study on the involvement of SJ and apico-basal polarity pathways in the Yki phenotypes (Robbins et al., 2014). The current work looks more convincing to me, but it would be nice to repeat some of the previous experiments to see if some phenotypes could have been previously missed. At the very least, this issue should be discussed and any clarifications regarding these apparently conflicting results would be helpful to include in the Discussion.

We are unclear on exact which data in the current report that the reviewer feels to be in conflict with our Robbins et al paper. One possible perceived discrepancy might be that in the initial report we stated that there were no obvious changes in the junctional or matrix markers that we examined. The current reports does not contradict this because in Robbins et al., we examined Coracle as a SJ maker, which is not strongly affected, but in this report we examined Kune, which is more strongly affected. Similarly, we had stained for the luminal marker Verm in Robbins et al, which is not affected, but in this report we describe effects on Serp. As stated in the text on pages 10-11, DrICE mutations do not equally affect the trafficking of all cargo. For the Robbins et al. paper, we stained mutants with the standard markers we had used in other papers, and those markers happen to not be affected by DrICE. When we found an association of DrICE with the clathrin and the endocytic system, we tested a wider panel of markers to see if any might be affected and identified ones that were. To clarify this potential confusion, we now note on pages 10-11 that the markers that were used in Robbins et al. are not mistrafficked.

Another possible perceived discrepancy is that in the Robbin et al. manuscript, we reported that polarity was not grossly affected in *DrICE* mutants. However, the localization and levels of Crbs were not examined in Robbins et al. In this manuscript we describe careful examination of the Crb phenotype, and find that overall polarity as assessed by Crb staining is grossly normal, but intracellular trafficking of Crb is affected at st. 14 in DrICE mutants.

Another possible perceived discrepancy could arise from our original report that SJ barrier function was intact, but in this manuscript we report changes in the levels of Kune, which is

required for barrier formation. These observations are not inconsistent because Kune levels are only reduced by 20% in *yki^{bs}* mutants, and localization of Mtf, Dlg and other SJ markers is intact *yki^{bs}* mutants. Loss of barrier function is typically accompanied by total junctional failure, with all SJ components spreading across the lateral membrane instead of being localized to the apical region of the lateral membrane^{4,5}. Thus, our findings that Dlg and Mtf have normal localization in a *yki^{bs}* mutant support the original observations that barrier function is intact in *yki^{bs}* mutants.

Reviewer #2

McSharry and Beitel provide strong genetic evidence for caspase activity that promotes endocytosis in the *Drosophila* tracheal system. The tracheal system is shown to be capable of apoptosis, but normally the caspase DrICE plays an alternate role. The role in endocytosis is evident from altered distributions of Crb in the trachea and epidermis of DrICE mutants, and with DrICE over-expression. DrICE loss of function also suppresses abnormal endocytosis that otherwise occurs in *yrt* mutants. Altered distributions of further markers are also shown in DrICE mutants. Full-length DrICE displays a diffuse cytosolic distribution, but antibodies for cleaved DrICE reveal a punctate pattern with enrichment to the apical domain and some co-localization with clathrin and Rab5. Yorkie and its transcriptional target Diap1 (an inhibitor of DrICE) both inhibit total levels of DrICE, as revealed by mutant analyses. Their mutants also display contorted dorsal trunks of the tracheal system, a defect that can be suppressed with a DrICE mutant or mimicked with DrICE overexpression but does not seem to be due to apoptosis. Through single and double mutant analyses, DrICE is shown to be responsible for a reduced deposition of luminal material in Yorkie mutants. Overall, a demonstration of a role for a caspase in endocytosis in a whole animal should be of general interest. However, a number of issues should be addressed.

We thank the reviewer for their support.

1. It was unclear how to interpret the comparisons of the aPKC and pSrc distributions in the *shrub* and DrICE mutants. *Shrub*, aPKC and pSrc seem to be indirectly associated with endocytosis. Can the authors clarify their rationales and conclusions in the manuscript?

The subcellular distributions of aPKC and pSrc in DrICE mutant embryos were shown because these two proteins have been shown to have roles in tracheal tube size control, and we believe they are mislocalized in DrICE mutants. The distributions of these markers were shown in *shrub* mutants because it had not previously been shown that disruptions of the endocytic pathway alter the subcellular distributions of aPKC and pSrc in the trachea. However, as the phenotypes of the aPKC and pSrc in DrICE embryos are not as strong as other markers, per the requests of this reviewer and reviewer 3, in the revised manuscript we removed discussion and data related to aPKC and pSrc, and focus on more strongly affected makers. We also removed the data for the other markers in a *shrub* mutant since those data have been previously published.

2. The authors claim that “pSrc is reduced at the apical membrane in both DrICE17 and *shrb4* mutant trachea (Fig. 3g-i)”, but the data are not convincing.

As discussed above, we removed aPKC and pSrc in the revised manuscript to focus on more strongly affected makers.

3. The colocalization of cleaved DrICE with clathrin and Rab5 was unconvincing. It seems that it could be coincidental.

To better assess the c-localization of DrICE^{CST9478} with endocytic markers, we have optimized the fixation and staining protocols for the various antibodies and quantified all colocalization. The association of DrICE^{CST9478} with clathrin is robust, with a Pearson's R-value of 0.25 and 23% overlap. Beyond the statistics, several aspects of the co-localization with clathrin are convincing:

First, if there were no significant association of DrICE^{CST9478} with clathrin, we would expect the correlation of DrICE^{CST9478} staining and clathrin to be close that for other endocytic markers. However, the R value for clathrin-DrICE^{CST9478} colocalization is 0.25, but for the other endocytic markers examined, the R-values are 0.13, 0.10 and 0.06, supporting the idea that there is in fact a non-random association of DrICE and clathrin. Note that correlation was done on individual confocal sections rather than projections, thus reducing the chance of specious overlap.

Second, clathrin staining does not form a continuous belt at the apical surface, leaving significant amounts of unstained apical area. Despite the clathrin-free areas, the majority of the apical DrICE^{CST9478} staining coincides with clathrin staining, indicating the colocalization of DrICE^{CST9478} and clathrin is not a coincidence (new Fig. 2I, yellow arrowheads).

Third, if there is an association between DrICE^{CST9478} and clathrin, then one would predict that DrICE^{CST9478} and clathrin should co-localize in the cytosol since clathrin mediates TGN-endosome and other intracellular trafficking as well as endocytosis. Consistent with this prediction, intracellular association between DrICE^{CST9478} staining and Clathrin is apparent (new Fig. 2I, inset).

Fourth, the significance of co-localization of DrICE^{CST9478} staining and clathrin is strongly supported by the observed changes in the intracellular trafficking of 4 tracheal size markers in DrICE mutants. If the co-localization of DrICE and clathrin was coincidental rather than functional, there would have to exist an alternative mechanism by which DrICE influences endocytic trafficking that does not involve association of DrICE with the endocytic system. Given these observations and mammalian data showing that caspases can cleave endocytic cargo and trafficking components^{6,7}, the most straightforward interpretation of the collective data is the DrICE associates with and acts on cargo and/or trafficking machinery on clathrin-coated vesicles.

To clarify these points, we revised Fig. 2 to quantify co-localization for all markers, added details of colocalization calculation to the Fig. 2 legend and revised the paragraph on page 8 describing the colocalization.

Also, there are many puncta of the cleaved DrICE through the cytosol that lack colocalization with any markers of the endosomal system tested. Could a marker for bulk endocytosis reveal more colocalization? This or other analyses are important for consolidating the endocytosis connection.

We have also noted that there are cytosolic puncta that are not associated with any tested markers for the endocytic system, but we are not attempting to make the case that all DrICE^{CST9478} associates with the endocytic system. For example, although full length DrICE is broadly distributed through the cytoplasm, it also has a fairly punctate appearance rather than uniform staining seen with cytosolic GFP. If DrICE becomes activated in these puncta, they would not necessarily colocalize with the endocytic system. Further, given the many roles of caspases, it is not clear that the *only* function of DrICE in tracheal morphogenesis is endocytic. For example, there are clearly DrICE puncta in the nucleus, which should not

contain any endocytic machinery. We propose that DrICE is a dynamic molecule that localizes to several processes or compartments and thus does not have complete co-localization with any single compartment. We believe that critical feature to consider is that at least some DrICE is specifically associating with the endocytic system, which is completely consistent with DrICE mutations affecting endocytic trafficking.

To clarify the point that there may be multiple pools of DrICE, we added the following sentence to the section of clathrin colocalization on page 8:

In addition, the incomplete complete co-localization of DrICE^{CST9478} staining with Clathrin suggests the multiple pools of DrICE exist that could support DrICE acting in multiple independent functions, including apoptosis.

Also, the degree of colocalization is important for the authors' model for a "mechanism by which tracheal cells avoid death in the face of activated caspases...[in which]...DrICE localizes to the endocytic system, which may help limit DrICE function to the endocytic pathway".

This is a valid point, but as mentioned above, since there is insufficient room in the manuscript to substantiate the details of the of a possible compartmentalization argument, we have removed reference to the compartmentalization model from the paper.

4. There are substantial puncta detected with the antibody for the cleaved DrICE even in a null mutant for DrICE that lacks staining for total DrICE. To what degree do these non-specific puncta colocalize with endocytic markers, or with Crumbs (relevant to Fig S1)?

The reviewer is correct that there is DrICE^{CST9478} staining even in *DrICE^{Δ1}* zygotic null embryos, however this is entirely consistent with the considerable maternal contribution of DrICE that is observed and quantified by Western blotting in old Fig. 1n-o (new Fig. 1o and Fig. S1i). To determine whether the residual staining in DrICE^{CST9478} embryos was non-specific staining, we examined late larval tissue that should have essentially no maternal DrICE signal. While we observed abundant DrICE^{CST9478} staining in wild-type wing discs, the staining was absent in the *DrICE^{Δ1}* mutant (new panels Fig. S2e, g). We clarify this point in the Figure 2 legend on page 15.

5. The last section of the results briefly describes analyses in other tissues. The section and the data are difficult to interpret because of the minimal explanations by the authors. Moreover, the investigations of the colocalization between the cleaved DrICE and the endocytic markers in the hindgut (Fig S3) have the issues raised above, but additionally lack any quantification.

We agree with the reviewer that this section was overly abbreviated in an effort to conform to word limits. Given that we are still constrained by word limits, the concerns from reviewer 3 and the request by the editor that we streamline the figures, we have opted to eliminate the analysis of other tissues.

6. A typo: in "over-expression of DrICE in the tracheal system increases Crb levels at stage 16 (p<0.0005, Supplementary Fig. 4a-c)", "Supplementary" should be deleted.

Corrected.

Reviewer #3

This is an interesting manuscript, which follows up on their discovery of a function for Yki and its target gene DIAP1 in restricting tracheal tube elongation by antagonising DrICE, which acts downstream of DIAP1 to promote elongation (PLoS One 2014 9(7): e101609). The new manuscript contains a great deal of data but most importantly reveals a physiological function for DrICE in promoting Crb endocytosis in trachea that is complementary to its normal role in promoting apoptotic cell death in other tissues (Fig 3a and 4a-c). The loss-of-function phenotype of DrICE mutants can be rescued by simultaneous mutation of Yurt, another protein that is known to promote Crb endocytosis (Fig 4h-l). Overall, these are important new findings that should be published in Nature Communications subject to the following major revisions.

We thank the reviewer for their support.

Major comments:

1. There are currently 6 highly complex figures that need to be reduced to 4 simple ones. Fig 1 is fine as is. Fig 2 should be placed in Supplementary Figures. Figs 3a and 4a-c should be combined into a new Fig 2 that is simple and emphasises the most significant finding of the manuscript - regulation of Crb by DrICE. The remainder of Fig 3 (3d-i and 3j-r) should be removed from the manuscript. The remainder of Fig 4 (4d-l) should become a simplified new Fig 3 that highlights only the rescue by yurt, leaving out extraneous data on Serp, Kune-Kune and Mtf (which are not in the least convincing). The current model Fig 6 then becomes the new Fig 4. Current Fig 5 should be removed from the manuscript.

Per the reviewer's and editor's request, we have streamlined the figures and text along the lines of the reviewer's suggestions. We have eliminated old figure 5 and extensively revised and simplified the new figures 3 and 4. We have also cropped and resized images to more clearly show the effects of DrICE on markers. The text of pages 8-10 has been revised accordingly.

2. The effect of DrICE on aPKC, p-Src, Serp and Uif was not convincing and should simply be removed from the manuscript.

We agree that changes to some of the markers presented in the original manuscript are less obvious, and thus have removed data for aPKC, pSrc, and Mtf from the manuscript. However, in addition to Crb, the markers Serp, Kune and UIF also show strong, statistically significant effects. As the other reviewers in general found the mis-trafficking data convincing, we have therefore retained the data for these markers.

Also, since the effects of DrICE on Dlg are subtle, we now use Dlg as a marker that is modestly by DrICE, which helps make the important point that markers are not equally regulated by DrICE. The text at the top of page 10 has been revised accordingly.

3. The effect of DrICE on Crb outside of the trachea (e.g.: Hindgut Fig S4 or Epidermis Fig 3j,k) seemed much less clear than in the trachea itself. I would suggest either removing this data from the manuscript or making a single separate supplementary figure that clarifies the tracheal-specific nature of this phenotype.

Per the reviewer's suggestion, we have eliminated the non-tracheal data in the paper.

4. The abstract needs to be amended to focus on Crb, and remove reference to 'at least seven ... proteins', as the data only support a clear effect on Crb (which could indirectly affect others).

We have revised the abstract to account for the removal of three markers from the manuscript. But we are reluctant to focus the entire abstract on Crb. Given that Crb levels and localization are not strongly affected at st. 16 when other markers are significantly affected, it seems premature to conclude that the trafficking defects in Kune, Serp, and UIF are secondary to defects in Crbs. We therefore took the more conservative approach in the abstract of stating what we have observed, which is that DrICE regulates "...endocytic trafficking of at least four cell polarity, cell junction and apical extracellular matrix proteins involved in tracheal tube size control: Crumbs, Uninflatable, Kune-Kune and Serpentine."

References

1. Baer, M.M. *et al.* The role of apoptosis in shaping the tracheal system in the Drosophila embryo. *Mechanisms of development* **127**, 28-35 (2010).
2. Xu, D. *et al.* The effector caspases drICE and dcp-1 have partially overlapping functions in the apoptotic pathway in Drosophila. *Cell death and differentiation* **13**, 1697-1706 (2006).
3. Amcheslavsky, A. *et al.* Plasma Membrane Localization of Apoptotic Caspases for Non-apoptotic Functions. *Developmental cell* **45**, 450-464 e453 (2018).
4. Lamb, R.S., Ward, R.E., Schweizer, L. & Fehon, R.G. Drosophila coracle, a member of the protein 4.1 superfamily, has essential structural functions in the septate junctions and developmental functions in embryonic and adult epithelial cells. *Molecular biology of the cell* **9**, 3505-3519 (1998).
5. Paul, S.M., Ternet, M., Salvaterra, P.M. & Beitel, G.J. The Na⁺/K⁺ ATPase is required for septate junction function and epithelial tube-size control in the Drosophila tracheal system. *Development* **130**, 4963-4974 (2003).
6. Zehendner, C.M., Librizzi, L., de Curtis, M., Kuhlmann, C.R. & Luhmann, H.J. Caspase-3 contributes to ZO-1 and Cl-5 tight-junction disruption in rapid anoxic neurovascular unit damage. *PloS one* **6**, e16760 (2011).
7. Duclos, C.M. *et al.* Caspase-mediated proteolysis of the sorting nexin 2 disrupts retromer assembly and potentiates Met/hepatocyte growth factor receptor signaling. *Cell Death Discov* **3**, 16100 (2017).

REVIEWERS' COMMENTS:

Reviewer #1 (Remarks to the Author):

The authors have addressed all my major questions and the revised ms clarifies all the key issues. Therefore, I recommend publication of this interesting ms at this time.

Hermann Steller

Reviewer #2 (Remarks to the Author):

The authors have effectively addressed my past concerns. Their paper's demonstration of an endocytic role of a caspase for tissue morphogenesis should be of general interest to readers of Nature Communications.

Reviewer #3 (Remarks to the Author):

I am satisfied with the revised manuscript.

Response to Reviewer Comments:

The reviewers did not have any comments, therefore there are no issues to address.